# Quercetin Improved Muscle Mass and Mitochondrial Content in a Murine Model of Cancer and Chemotherapy-Induced Cachexia

**DOI:** 10.3390/nu15010102

**Published:** 2022-12-25

**Authors:** Brandon N. VanderVeen, Thomas D. Cardaci, Patrice Cunningham, Sierra J. McDonald, Brooke M. Bullard, Daping Fan, E. Angela Murphy, Kandy T. Velázquez

**Affiliations:** 1Department of Pathology, Microbiology and Immunology, School of Medicine, University of South Carolina, Columbia, SC 29209, USA; 2AcePre, LLC, Columbia, SC 29209, USA; 3Department of Cell Biology and Anatomy, School of Medicine, University of South Carolina, Columbia, SC 29209, USA

**Keywords:** cancer, chemotherapy, cachexia, skeletal muscle, mitochondria, complementary medicine, nutraceuticals

## Abstract

A cachexia diagnosis is associated with a doubling in hospital stay and increased healthcare cost for cancer patients and most cachectic patients do not survive treatment. Unfortunately, complexity in treating cachexia is amplified by both the underlying malignancy and the anti-cancer therapy which can independently promote cachexia. Quercetin, an organic polyphenolic flavonoid, has demonstrated anti-inflammatory and antioxidant properties with promise in protecting against cancer and chemotherapy-induced dysfunction; however, whether quercetin is efficacious in maintaining muscle mass in tumor-bearing animals receiving chemotherapy has not been investigated. C26 tumor-bearing mice were given 5-fluorouracil (5FU; 30 mg/kg of lean mass i.p.) concomitant with quercetin (Quer; 50 mg/kg of body weight via oral gavage) or vehicle. Both C26 + 5FU and C26 + 5FU + Quer had similar body weight loss; however, muscle mass and cross-sectional area was greater in C26 + 5FU + Quer compared to C26 + 5FU. Additionally, C26 + 5FU + Quer had a greater number and larger intermyofibrillar mitochondria with increased relative protein expression of mitochondrial complexes V, III, and II as well as cytochrome c expression. C26 + 5FU + Quer also had increased MFN1 and reduced FIS1 relative protein expression without apparent benefits to muscle inflammatory signaling. Our data suggest that quercetin protected against cancer and chemotherapy-induced muscle mass loss through improving mitochondrial homeostatic balance.

## 1. Introduction

Cachexia, the unintentional loss of body weight with chronic disease, is prevalent in chronic obstructive pulmonary disease, heart failure, chronic kidney disease, and multiple cancers [1,2]. Cachexia is associated with reduced physical function, decreased tolerance for treatment, and increased mortality. A cachexia diagnosis is consistent with a doubling in duration of hospital stay and an additional $4000/case compared to non-cachectic patients [1,2]. Indeed, ~80% of cancer patients with a cachexia diagnosis do not survive treatment and 20% of all cancer-related deaths can be attributed to cachexia [3]. Additionally, the loss of lean mass with cachexia impairs chemotherapy treatment tolerance and can exacerbate functional decrements leading to further functional dependencies and accrued health care cost [4,5,6]. Both the underlying cancer and its treatment can negatively impact muscle quality and mass, exacerbating cachexia’s deleterious impact.

While our understanding of chemotherapy-induced cachexia is still in its earliest mechanistic stages, it appears that the muscle mitochondria sit at a nexus of cancer and chemotherapy-induced skeletal muscle dysfunction [7,8,9,10]. Preclinical models of cancer-cachexia have demonstrated losses in mitochondrial content, biogenesis, dynamics, and function concomitant with reduced muscle function [7,11,12,13,14]. More recently, cytotoxic chemotherapies, including doxorubicin and 5-fluorouracil (5FU) were demonstrated to impair mitochondrial function, and decrease content without the presence of the tumor environment [9,10]. Recently, new models examining the combination of 5FU containing therapies with the colon-26 (C26) colorectal cancer model of cancer cachexia has demonstrated indices of cachexia even with stunted tumor growth [15]. Furthermore, they showed these deficits can be improved with a dietary intervention that did not impair the anti-cancer efficacy of chemotherapies [15].

While new therapeutics continue to be tested each year, unfortunately, there are currently no approved therapies for cancer cachexia. Although traditional nutritional support (i.e., hypercaloric diets) has not been effective in preventing cancer associated weight loss, specialized anti-inflammatory and antioxidant diets as well as nutraceuticals continue to be a safe alternative to pharmacological intervention with therapeutic promise [15]. Quercetin (3,4,5,7-pentahydroxylflavone) is an organic polyphenolic flavonoid commonly found in fruits and vegetables, such as grapes, apples, blueberries, and onions [16,17]. Pre-clinical studies have found it to demonstrate anti-inflammatory, antiviral, antioxidant, cardio-protective, anti-carcinogenic, and neuroprotective properties, among many others [18,19,20,21,22,23,24,25,26,27,28]. These properties associated with quercetin have established it as a dietary agent or complementary medicine with potential to treat inflammatory diseases and cancer, as well as related conditions like cancer cachexia [20,27,29,30,31].

We recently conducted a sub-chronic quercetin toxicity study in CD2F1 mice and found that quercetin had no deleterious effects or toxicities at various dosages [32]. The purpose of the current study was to examine quercetin’s efficacy in preventing cachexia in C26 tumor-bearing mice treated with 5FU. We hypothesized that C26 tumor-bearing mice treated with 5FU would experience significant body weight and muscle mass loss with disrupted mitochondrial homeostasis; however, supplementing with 50 mg/kg quercetin as a complementary therapy with 5FU would stave off cachexia’s progression. We found that C26 tumor-bearing mice given 5FU and quercetin had partially improved muscle mass, preserved myofibrillar cross section area (CSA), and protected muscle mitochondrial content compared to C26 tumor-bearing mice given 5FU alone.

## 2. Materials and Methods

### 2.1. Animals

Male (n = 20) *CD2F1* (CD2F1) hybrid mice were purchased from Charles River Laboratory (Raleigh, NC, USA) at 10 weeks of age. Mice arrived at our facilities and were given AIN76 diet and allowed to acclimate to the new facilities and purified diet for 4 weeks. Mice were kept in ventilated cages (5 per cage), on a 12:12 h light/dark cycle, in a humidity and temperature control room (~22 °C) and given ad libitum access to water and food. Animal handling and experiments were performed to minimize pain and discomfort. At 14 weeks of age mice were separated into four groups following stratified random sampling for body weight and lean mass: (1) Control (n = 5), (2) C26 (n = 5), (3) C26 + 5FU (n = 5), (4) C26 + 5FU + Quer (n = 5). Control mice were non-tumor-bearing mice given PBS i.p. and propylene glycol p.o. as vehicle controls. Body weights were measured daily. Food weights per cage were measured daily and average food intake was calculated by the difference in food weight between days. After 10 days of C26 tumor growth, both 5FU and quercetin were given daily for 5 days until Day 15. Mice were euthanized 24 h following the last 5FU injection (Day 16). Hindlimb muscles and select organs were excised, weighed, and snap frozen in liquid nitrogen while mice were under sedation (2% Isoflurane, 2 L/min O_2_). All procedures involving animals were reviewed and approved by the Institutional Animal Care and Usage Committee (IACUC) at the University of South Carolina in accordance with the American Association for Laboratory Animal Science.

### 2.2. Cell Culture and Cell Implantation

C26 (colon 26 adenocarcinoma) cells were gifted from Dr. Andrea Bonetto’s laboratory and grown in complete DMEM (10% fetal bovine serum, 1% penicillin/streptomycin). Cells were split at 70% confluency and passage 3 was used for cell implantation. On Day 0, CD2F1 mice were given a subcutaneous injection of 1 × 10^6^ C26 cells in the subscapular area as previously described [33]. Tumors were palpable at Day 8 in all C26 tumor-bearing mice.

### 2.3. Chemotherapy Intervention

Fluorouracil (5FU; VWR; CAS#:51–21-8) was solubilized in warmed phosphate-buffered saline (PBS) at 3.0 mg/mL, sterile filtered with a 0.2 µm filter and administered at the beginning of the light cycle (0700) via i.p. injection to mice with established tumor (day 11 after C26 implantation). We have previously demonstrated C57BL/6 mice are susceptible to 5FU-induced cachexia at a dose of 40 mg/kg of lean mass (LM) [34,35]; however, CD2F1 mice were more sensitive to 5FU (data not shown) and therefore given 5FU at 30 mg/kg of LM.

### 2.4. Quercetin Administration

Quercetin (Sigma-Aldrich, St. Louis, MO, USA; catalog#: Q4951) was made daily in propylene glycol at 4 mg/mL, concentration at which to not exceed maximum volume for oral gavage. Mice were given either quercetin at 50 mg/kg body weight (BW) or vehicle (propylene glycol) p.o. on Day 11 after C26 implantation at the end of the light cycle (1900).

### 2.5. Dual-Energy X-Ray Absorptiometry

Mice were subjected to Dual Energy X-ray Absorptiometry (DEXA; Lunar PIXImus; General Electric, Boston, MA, USA) and images were analyzed to assess lean body mass (LM) [34].

### 2.6. Muscle Histology

Skeletal muscle histology was assessed via hematoxylin and eosin (H&E) staining on tibialis anterior (TA) muscle cryosections. The TA muscle was dissected, snap-frozen in liquid nitrogen with Optimum Cutting Temperature (OCT) embedding medium placed on the most distal portion of the TA and stored in −80 °C. Tissue was acclimated to the cryostat temperature (−24 °C) prior to collection of transverse sections (10 μm) on Fisherbrand Superfrost Plus Microscope Slides. TA tissue slides were then stored at −80 °C until staining. Slides were fixed in ice-cold acetone and H&E staining was completed as previously described [34,36]. Images of H&E stains were taken at 20× and 40× magnification using a Keyence BZX800 microscope. Muscle cross sectional area (mCSA) was measured using ImageJ by measuring the circumference of 500–700 myofibers/mouse (n = 3–5/group).

### 2.7. Myofiber Metabolic Phenotype

To assess skeletal muscle myofiber metabolic phenotype succinate dehydrogenase (SDH) activity was measured. SDH is an enzyme bound to the inner mitochondrial membrane and SDH activity is higher in type 1 fibers [37]. TA tissue cryosections were taken out of the −80 °C and air dried on the laboratory bench and then incubated in an SDH staining solution containing 0.5 mg/mL nitroblue tetrazoliumand (Sigma, Cat#: N5514) and 50 mM sodium succinate (Sigma-Aldrich, St. Louis, MO, USA Cat#: S2378) in PBS for 30 min at 37 °C [38]. Slides were then washed in distilled water 3 × 1 min before mounting with warmed glycerol mounting media and cover slipped. Images of SDH stains were taken at 4×, 20×, and 40× using a Keyence BZX800 microscope. SDH activity was analyzed by measuring the circumference and mean grey value of ~300 fibers from throughout the entire muscle. Images were inverted thus higher values corresponded with higher SDH activity.

### 2.8. Electron Microscopy

Mitochondrial number and size were assessed in the soleus muscle using transmission electron microscopy (TEM). Skeletal muscle was prepared for TEM as previously described [36]. The soleus was excised and initially fixed in 2.5% EM-grade glutaraldehyde followed by a secondary fixation in 1% osmium tetroxide and 1.5% K+ ferricyanide. After dehydration with increasing concentrations of ethanol, the samples were transferred to acetonitrile and embedded in Polybed 812. Ultrathin (~70 nm) sections were cut with a Leica Reichert Ultracut R ultramicrotome (Leica Microsystems GmbH, Wetzlar, Germany), collected on copper grids and stained with uranyl acetate and lead citrate. A JEOL 1400+ transmission electron microscope (JEOL USA, Peabody, MA) was used to view sections, and digital images were obtained with an Advanced Microscopy Techniques XR 81 Camera (Advanced Microscopy Techniques, Woburn, MA). Tissues were searched for intact and aligned sarcomeres by an investigator blinded to the experimental groups. For each mouse (n = 3/group), ≥5 images were taken at 6000× across 3–5 myofibers within a single sagittal section. Mitochondrial number was taken by counting the number of discernible mitochondria within the field of view (F.O.V.) and averaged across the 5–7 images within a single mouse. Mitochondrial size was calculated by circling discernible mitochondria with ImageJ within the F.O.V. and averaged across the multiple images within a single mouse. Representative 1500× images were taken to further demonstrate distinguishable changes in mitochondrial number and size.

### 2.9. Western Blotting

Protein isolation and Western blotting was completed similar to what has been previously described [39]. The rectus femoris (RF) was teased from the other quadriceps muscles, cut in half (proximal/distal), snap frozen in liquid nitrogen, and stored at −80 °C. The distal RF was homogenized in Mueller Buffer (50 mM Hepes, 0.1% Triton-x, 4 mM EGTA, 10 mM EDTA, 15 mM Na_4_P_2_O_7_, 100 mM β-glycerophosphate, 25 mM NaF, and 5 mM NaVO_4_, with protease inhibitor cocktail) using a bead homogenizer at 4 °C. Samples were centrifuged and the supernatant was collected and diluted with a diluent buffer (50% Glycerol, 50 mM Na_4_P_2_O_7_, 2.5 mM EGTA, and 1 mM β-mercaptoethanol with protease inhibitor cocktail). Protein concentrations were determined using the Bradford standard protein assay and protein integrity was confirmed with SDS-PAGE and amido black staining. Between 10–50 ug of muscle protein homogenate was separated via SDS-PAGE and transferred to a PVDF membrane. Membranes were stained with Ponceau Red to confirm even loading and transfer efficiency. Membranes were then washed with tris buffered saline with 0.1% Tween (TBST) before blocking for 1 h at room temp with 5% milk TBST. Membranes were then incubated in primary antibody (Table 1) overnight at 4 °C under gentle agitation. Membranes were then washed 3 × 5 mins and incubated with the appropriate (anti-mouse or rabbit) HRP-linked secondary in 5% milk TBST for 1 h at room temperature. After a final 3 × 5 min wash membranes were incubated in chemiluminescent HRP substrate before visualization with a Syngene G:box. Images were then scanned, and relative protein expression was determined with ImageJ. Investigator was blinded to sample loading scheme and images were cropped after analysis for representation. All samples (full Ns) were run together for each protein. Dotted lines indicate where gels were cropped.

### 2.10. Real Time—Polymerase Chain Reaction (RT-PCR)

RNA from the proximal portion of the RF was isolated as previously described [36]. Briefly, RNA was extracted from the RF using the TRIzol/isopropanol/chloroform procedure (Life Technologies, Gibco-BRL, Carlsbad, CA). RNA sample quality and quantities were verified using a Nanodrop One Microvolume UV-Vis Spectrophotometer (Thermo Fisher Scientific, Waltham, MA). Samples with A260/A280 and A260/A230 values >1.8 were used for cDNA synthesis using High-capacity Reverse Transcriptase kit (Applied Biosystems, Foster City, CA). Quantitative RT-PCR analysis was carried out as per the manufacturer’s instructions (Applied Biosystems) using Taq-Man Gene Expression Assays. Data were normalized to vehicle-treated controls and compared with five reference targets (B2M, TBP, HPRT, 18 s, and H2AFV), which were evaluated for expression stability using the GeNorm algorithm.

### 2.11. Plasma Interleukin-6 Analysis

Blood was collected retro orbitally prior to euthanasia while the mice were anesthetized. Whole blood was collected and put in an EDTA-coated tubes and store on ice prior to centrifugation (3000× *g* at 4 °C for 10 min) for plasma collection. A commercially available IL-6 enzyme-linked immunosorbent assay (ELISA) kit was obtained from BioLegend (San Diego, CA, USA). Briefly, a Costar clear 96-well plate (Corning, NY, USA) was coated with IL-6 capture antibody and allowed to incubate overnight. The plate was then blocked with assay diluent buffer, and IL-6 standards and plasma samples were added to the plate. Wells were then washed and detection antibody was incubated in each well for 1 h. The wells were again washed and then incubated with avidin horseradish peroxidase reagent. After several washes, 3,3′,5,5′-tetramethylbenzidine substrate was added, and the reaction was developed for 20 min. The reaction was stopped with sulfuric acid, and absorbance was measured.

### 2.12. Statistical Analyses

Data were analyzed using Prism 8 statistical software (GraphPad Software, CA, USA). Unpaired *t*-test was used to determine differences between C26 + 5FU and C26 + 5FU + Quer. The healthy controls are shown as a reference point but were not included in statistical analysis. Additionally, the C26 alone group was not included in analysis, but is shown as a reference point for tumor growth without chemotherapy intervention. Data are presented as the mean ± SEM and the level of significance was set up at *p* ≤ 0.05.

## 3. Results

### 3.1. Animal Characteristics

At 14 weeks of age, male CD2F1 mice were injected with 1 × 10^6^ C26 cells (Day 0) and body weights were tracked 16 days. At Day 11, 3d after initial tumor palpation, mice were given daily i.p. injections of 5FU at 30 mg/kg of lean mass along with either propylene glycol or 50 mg/kg of quercetin (Figure 1A). C26 mice did not survive to Day 16, thus tumor and tissue weights were collected at Day 13. In mice given 5FU, acute administration of quercetin had no apparent impact on body weight loss (Figure 1A), nor did it impact 5FU’s tumor suppressive capabilities (Figure 1B). Quercetin did, however, spare RF muscle weight (Figure 1C), and gonadal fat pad weight (Figure 1D). C26 + 5FU + Quer mice had 14.4% greater RF weight (*p* = 0.02) compared to C26 + 5FU mice (Figure 1C). C26 + 5FU + Quer mice had 50.0% greater gonadal fat pad weight (*p* = 0.04) compared to C26 + 5FU (Figure 1D). On average, C26 + 5FU mice consumed 1.05 g of food/d meanwhile C26 + 5FU + Quer ingested 0.94 g of food/d. Non-tumor bearing control mice consumed an average of 2.94 g of food/d.

### 3.2. Quercetin Maintained Skeletal Muscle Cross Sectional Area

We then sought to further examine the anti-cachectic effects of quercetin. C26 + 5FU + Quer mice had 13.9% greater gastrocnemius weight (*p* = 0.055) and 23.7% greater EDL weight (*p* = 0.039) compared to C26 + 5FU mice (Figure 2A). While the other measured muscle weights did not achieve statistically significantly differences (*p* = 0.09*–*0.15), we sought to improve our sensitivity of detecting changes in muscle size by examining CSA. There was a 32.0% increase in TA mean CSA in C26 + 5FU + Quer compared to C26 + 5FU (Figure 2B,C). This can also be observed by a rightward shift in myofibrillar fiber size distribution in C26 + 5FU + Quer compared to C26 + 5FU (Figure 2D). We then sought to determine if there was a preferential sparing of glycolytic or oxidative fibers, and observed an increase in CSA across glycolytic, mixed, and oxidative fibers according to SDH activity (Figure 2E,G). Again, this can also be appreciated by examining the relationshi*p* between CSA and SDH activity demonstrating that C26 + 5FU + Quer had increased mCSA and SDH across the spectrum of fibers analyzed (Figure 2F).

### 3.3. Quercetin Maintained Skeletal Muscle Mitochondrial Size and Number

Quercetin has been hypothesized to protect against skeletal muscle wasting through improved oxidative metabolism and mitochondrial homeostasis. Therefore, we examined the impact of quercetin on mitochondrial size and number by TEM (Figure 3A). C26 + 5FU + Quer had 89.6% greater number of mitochondria (*p* = 0.04; Figure 3B) and 45.3% greater mitochondrial size (*p* > 0.01; Figure 3C).

### 3.4. Quercetin Improved Mitochondrial Content-Associated Skeletal Muscle Proteins

To further support the TEM results, we examined mitochondrial content proteins in whole muscle homogenates (Figure 4A). C26 + 5FU + Quer had increased relative expression of complex V (43.9%; *p* = 0.036; Figure 4B), III (237%; *p* = 0.024; Figure 4B), and II (270%; *p* < 0.01; Figure 4B) and a trending increase in complex I (618%; *p* = 0.08; Figure 4B) with no apparent changes to complex IV (8.3%; *p* = 0.44; Figure 4F) when compared to C26 + 5FU. Additionally, C26 + 5FU + Quer had increased relative expression of voltage dependent anion channel (VDAC) by 62.8% (*p* = 0.055; Figure 4G) and increased expression of cytochrome c (Cyto c) by 48.9% (*p* = 0.01; Figure 4H).

### 3.5. Quercetin Impacted Mitophagy-Associated Skeletal Muscle Proteins

We then investigated the mechanisms by which quercetin improves muscle mitochondrial content. First, we examined proteins associated with autophagy and particularly mitochondrial associated autophagy (mitophagy; Figure 5A). There were no observed differences in general autophagy proteins, P62 (*−*33.4%; *p* = 0.20; Figure 5B) and LC3b (*−*33.4%; *p* = 0.2; Figure 5C). Examination of mitophagy specific proteins showed a trend to increase Parkin expression by 10.2% (*p* = 0.10; Figure 5D) and a significant 555% increase in BNIP3 expression (*p* = 0.024; Figure 5E) in C26 + 5FU + Quer compared to C26 + 5FU.

### 3.6. Quercetin Differentially Impacted Mitochondrial Biogenesis, Fission, and Fusion

We further sought to investigate the mechanisms by which quercetin improves muscle mitochondrial content by examining mitochondrial dynamics proteins related to fission and fusion (Figure 6A). C26 + 5FU + Quer had increased relative expression of MFN1 (33.3%; *p* = 0.003; Figure 6B) without any apparent changes in MFN2 (11.1%; *p* = 0.25; Figure 6C). Additionally, there was a trend to increase OPA1 (82.9%; *p* = 0.06; Figure 6D). C26 + 5FU + Quer had reduced relative expression of FIS1 (*−*23.4%; *p* = 0.02; Figure 6E) compared to C26 + 5FU without any apparent changes to DRP1 (18.9%; *p* = 0.16; Figure 6F). C26 + 5FU + Quer had reduced relative expression of TFAM (*−*47.5%; *p* = 0.03; Figure 6G). C26 + 5FU + Quer also had reduced gene expression of PPARG (*−*64.0%; *p* = 0.04; Figure 6H) compared to C26 + 5FU without changes in PPARGCA1 gene expression (10.3%; *p* = 0.36; Figure 6.

### 3.7. Quercetin Has a Modest Impact on Skeletal Muscle Inflammatory Signaling

Quercetin has also been suggested to contain anti-inflammatory effects which are thought to contribute to its anti-cachectic properties. Therefore, we examined the impact of quercetin on plasma IL-6 (Figure 7A) and skeletal muscle inflammatory signaling proteins, STAT3, P65/NFκβ, and P38/MAPK (Figure 7B). It did not appear that quercetin decreased plasma IL-6, phospho:total (p/t) STAT3 (33.2%; *p* = 0.36; Figure 7C) or P38 (−30.0%; *p* = 0.19; Figure 7D); however, C26 + 5FU + Quer had increased p/tP65 (41.9%; *p* = 0.03; Figure 7E).

## 4. Discussion

Quercetin has been posed as a possible treatment of many inflammatory conditions, given its multi modal effects [18,19,20] and its excellent safety profile. In the current study we sought to investigate quercetin’s ability to protect against skeletal muscle mass loss in a novel model of cancer and chemotherapy-induced cachexia. We recently showed quercetin was safe at multiple doses [32] and now show that quercetin partially preserved muscle mass and protected muscle mitochondrial content and quality control in the context of cancer and chemotherapy.

Skeletal muscle mitochondria appear central to cancer and chemotherapy-induced cachexia [7,9,10,14,40,41,42,43]. Quercetin has been shown to improve mitochondrial content and/or quality in several tissues [44,45,46,47,48]. Skeletal muscle mitochondrial content (demonstrated by mitochondria associated proteins Complex I-V, VDAC, and Cyto C) is primarily regulated by the balance of mitochondrial biogenesis (coordinated by PGC1-α and PPARγ) and autophagy/mitophagy (coordinated by LC3, P62, Parkin, BNIP3), which involves the fusion of new or existing mitochondria to the mitochondrial network (coordinated by MFN1, MNF2, and OPA1) or the fission of damaged or old mitochondria (coordinated by FIS1 and DRP1). Mitochondrial function is often defined by the magnitude of ATP production in concert with H_2_O_2_ or free radical production. If ‘dysfunctional’, the mitochondria should be split and tagged by E3 ligase Parkin or apoptotic BNIP3 for removal by mitophagy [49,50,51]. Disruptions to these many processes have been demonstrated to disrupt skeletal muscle mass maintenance and function [51,52]. In the current study we show that quercetin increased mitochondrial number and size assessed via TEM and increased overall content assessed via Western blot. Additionally, we propose that this occurred through increased mitochondrial fusion and decreased fission.

In general, acute quercetin administration equilibrated the molecular signaling to protect against mitochondrial dysfunction. Our data show that quercetin increased BNIP3 expression and decreased TFAM protein and *PPARG* gene expression when compared to vehicle treated C26 + 5FU mice. The increase of TFAM protein and *PPARG* gene expression observed in C26 + 5FU mice might be a compensation due to the loss of mitochondrial content which is not apparent in the quercetin treated mice. Therefore, the compensatory upregulation of markers associated with mitochondrial biogenesis would not need to occur in quercetin treated mice. Similarly, the loss of BNIP3 in C26 + 5FU mice could be due to a loss of mitochondrial content in which the mitochondrial specific BNIP3 would be low in a whole muscle homogenate. BNIP3 is thought to be located on the outer mitochondrial membrane, while Parkin is located in the sarcoplasm and is recruited to the mitochondria, which helps explain why BNIP3 was reduced while Parkin was not changed [51,53]. This is further supported by our results with general autophagy markers, P62 and LC3. However, further examination of isolated mitochondria is necessary to support this conclusion.

Quercetin has been shown to improve muscle mass in cancer and rheumatoid arthritis preclinical models [20,31,54,55,56,57,58]. Our group previously showed that 25 mg/kg of quercetin preserved muscle mass and strength in the *Apc^Min/+^* model of intestinal cancer [20]. Quercetin (35 mg/kg) also protected against muscle mass loss in the C26 model of cachexia [31]. It was hypothesized that this sparing effect was due to a decrease in plasma IL-6 which has been shown as a key driver of muscle mass loss and muscle mitochondrial loss in the *Apc^Min/+^* [59,60,61]. Differences observed in the inflammatory signaling might be due to the acute administration (5 oral gavages) of quercetin in the current study. We show that quercetin did not reduce circulating IL-6 or muscle inflammatory signaling which are both known regulators of skeletal muscle mass with cachexia [62], and mitochondrial homeostasis balance can be modulated by inflammatory processes [7,59]. We then speculate that the improved mitochondrial control observed in the current study is more likely due to the antioxidant properties of quercetin rather than its anti-inflammatory properties. However, more studies are needed to determine if preventive and/or chronic administration of quercetin in tumor-bearing animals receiving chemotherapy can not only protect muscle mass but also maintain muscle function.

While the pursuit of novel pharmaceuticals continues in the effort to provide patients relief from cachexia’s burden, complementary therapies including plant-based medicines and exercise provide safe and promising results on sparing muscle mass and function [63]. In the current study we provide evidence in support of quercetin’s anti-cachectic properties demonstrated by preserved muscle mass and CSA with improved mitochondrial content. While quercetin has been previously shown separately to improve cancer cachexia [20,31,55,57] and chemotherapy-induced fatigue [30], we are the first to demonstrate that quercetin was efficacious in a combined model of cancer and chemotherapy-induced skeletal muscle atrophy. Certainly, future work is needed to best determine quercetin’s utility in treating cachexia in cancer patients.

## Figures and Tables

**Figure 1 nutrients-15-00102-f001:**
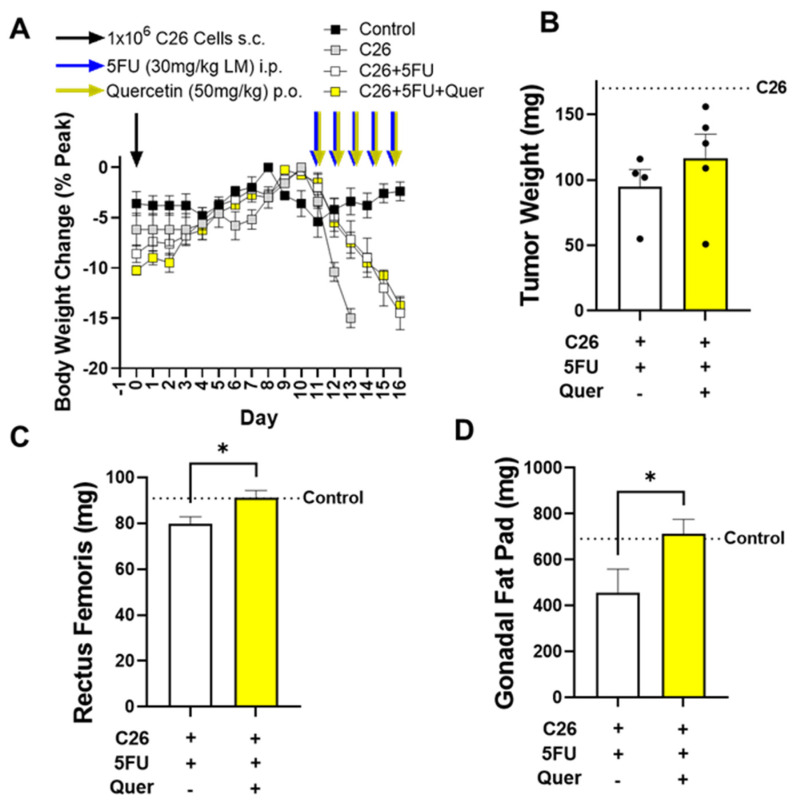
Quercetin’s impact on body and tissue weights. (**A**) Body weights shown as a percent of each mouse’s peak body weight over the course of the study. Mice were injected with 1 × 10^6^ C26 cells subcutaneously (s.c.) at Day 0. Mice were then given 5FU at 30 mg/kg of lean mass (LM) via intraperitoneal (i.p.) injection starting at Day 11. Mice were also administered either quercetin (n = 5) at 50 mg/kg body weight or propylene glycol (n = 5) via oral gavage (per os; p.o.) starting at Day 11. (**B**) Tumor weights in milligrams (mg) taken at euthanasia. An additional cohort of mice given C26 without 5FU are shown as reference point (dotted line) for tumor growth without intervention. (**C**) Rectus femoris and (**D**) gonadal fat pad weight in mg taken at euthanasia. n = 5/group. Unpaired *t*-test between C26 + 5FU and C26 + 5FU + Quer. * *p* ≤ 0.05.

**Figure 2 nutrients-15-00102-f002:**
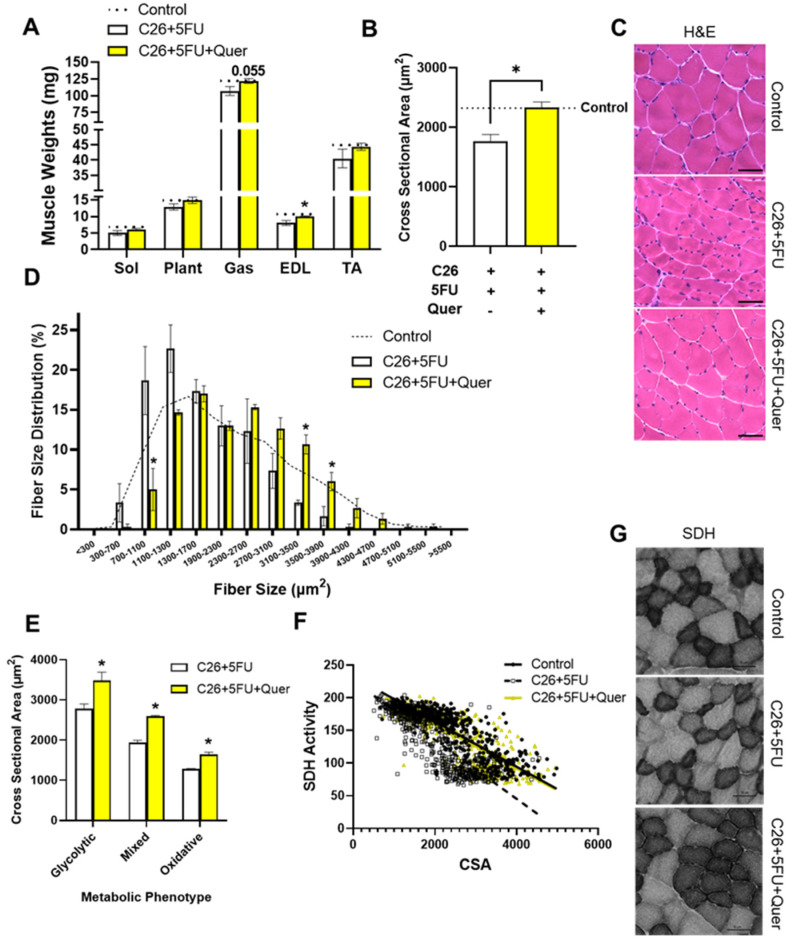
Quercetin increased muscle cross sectional area and succinate dehydrogenase activity. (**A**) Hindlimb muscle weights given in milligrams (mg)*—*soleus (Sol), plantaris (Plant), gastrocnemius (Gas), extensor digitorum longus (EDL), and tibialis anterior (TA). *(***B**) Mean myofibrillar cross sectional area (CSA) in micrometers^2^ (µm^2^) obtained from *(***C**) hematoxylin and eosin stained (H&E) TAs. *(***D**) myofibrillar size distribution given in relative percent across increasing fiber sizes (µm^2^). *(***E**) Mean CSA in µm^2^ across glycolytic, mixed, or oxidative fiber types obtained from succinate dehydrogenase (SDH) stained TAs. *(***F**) plotted linear relationshi*p* between single myofiber SDH activity and fiber size. *(***G**) Representative SDH images. Scale bar*—*50 µm. Unpaired *t*-test between C26 + 5FU and C26 + 5FU + Quer. n = 3*–*5/group. * *p* ≤ 0.05.

**Figure 3 nutrients-15-00102-f003:**
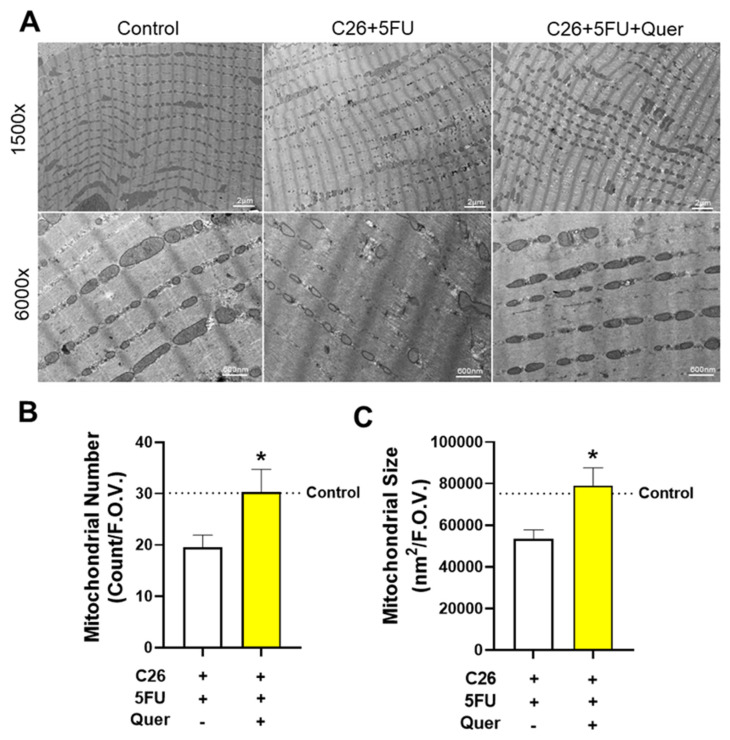
Quercetin increased mitochondrial size and number. (**A**) Representative transmission electron microscopy images of the soleus muscle taken at 1500× and 6000×. (**B**) Average number of mitochondria per field of view (F.O.V.) from 6000× images. (**C**) Average size in nanometers^2^ (nm^2^) of mitochondria per F.O.V. n = 3/group. 1500× scale bar*—*2 µm. 6000× scale bar*—*600 nm. Unpaired *t*-test between C26 + 5FU and C26 + 5FU + Quer. n = 3/group * *p* ≤ 0.05.

**Figure 4 nutrients-15-00102-f004:**
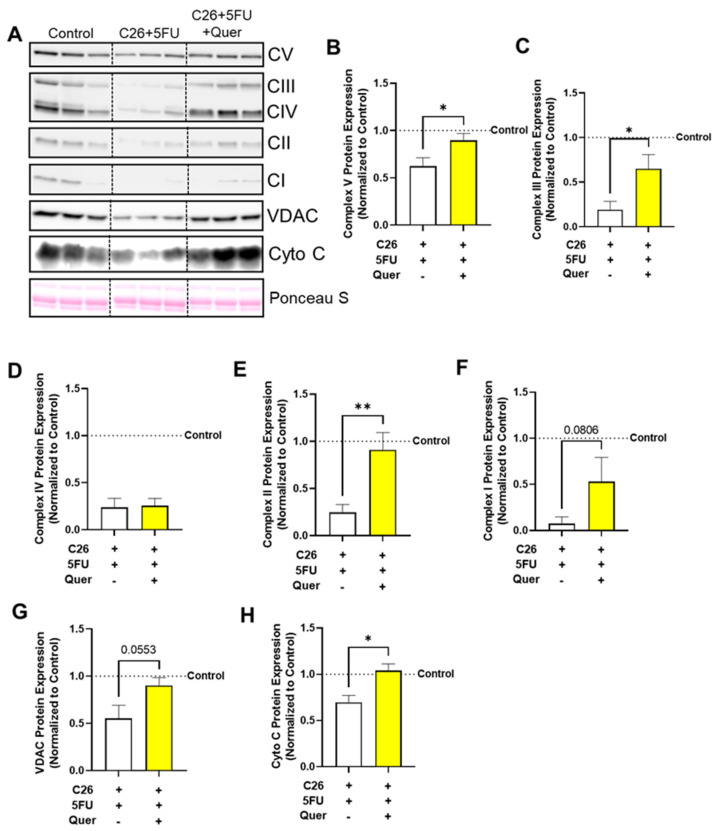
Quercetin increased mitochondrial content proteins. (**A**) Representative Western blots of mitochondrial complexes I-V, voltage dependent anion channel (VDAC), cytochrome c (Cyto C), and loading control Ponceau S from the rectus femoris muscle. Vertical dotted line demonstrates where images were cropped for representation. All samples were run on the same gel/membrane. Quantified relative expression of (**B**) Complex V, (**C**) Complex III, (**D**) Complex IV, (**E**) Complex II, (**F**) Complex I, (**G**) VDAC, and (**H**) Cyto C. Unpaired *t*-test between C26 + 5FU and C26 + 5FU + Quer. n = 3*–*5/group. * *p* ≤ 0.05, ** *p* < 0.01.

**Figure 5 nutrients-15-00102-f005:**
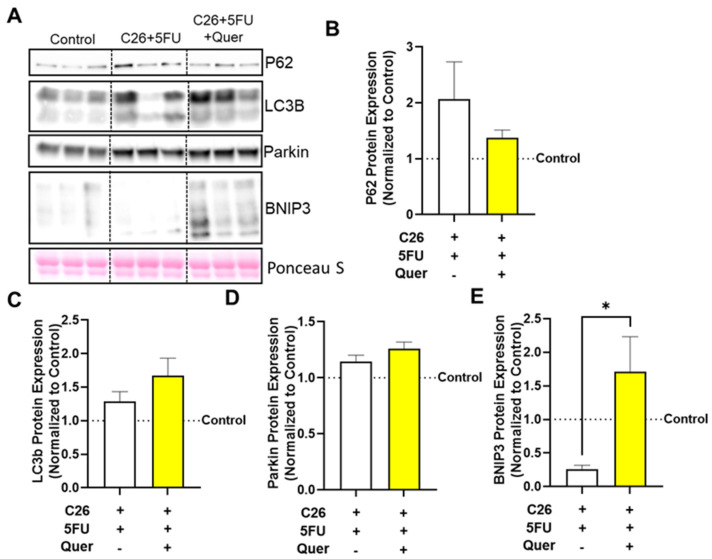
Quercetin has modest impacts of general autophagy and mitophagy proteins. (**A**) Representative Western blots of P62, light chain 3 (LC3B), Parkin, Bcl-2 interacting protein 3 (BNIP3), and loading control Ponceau S from the rectus femoris muscle. Vertical dotted line demonstrates where images were cropped for representation. All samples were run on the same gel/membrane. Quantified relative expression of (**B**) P62, (**C**) LC3B, (**D**) Parkin, and (**E**) BNIP3. Unpaired *t*-test between C26 + 5FU and C26 + 5FU + Quer. n = 3*–*5/group. * *p* ≤ 0.05.

**Figure 6 nutrients-15-00102-f006:**
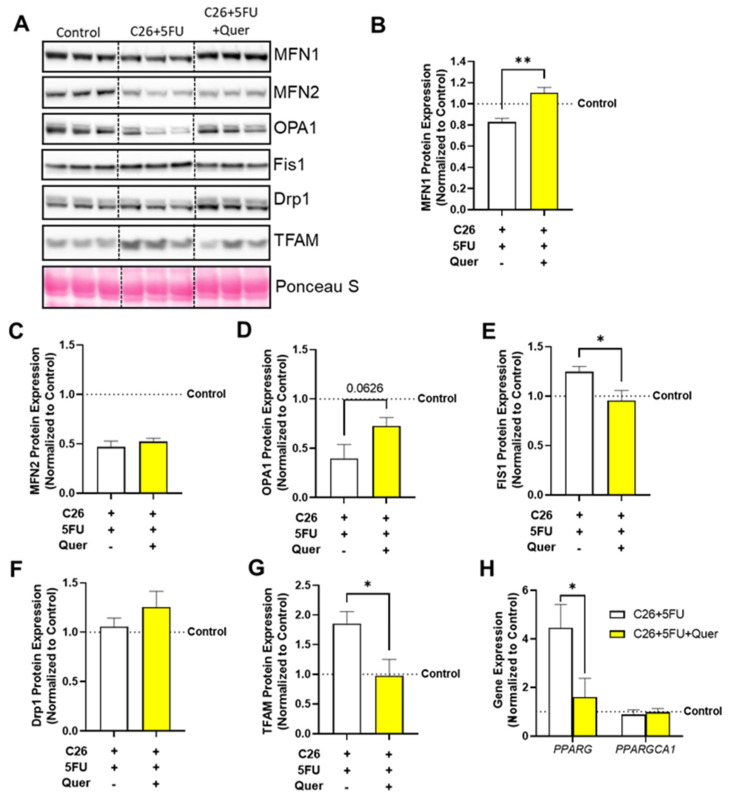
Quercetin normalizes mitochondrial dynamics proteins. (**A**) Representative Western blots of mitofusion (MFN) 1 and 2, optic atrophy 1 (OPA1), mitochondrial fission 1 (FIS1), dynamin-related protein 1 (DRP1), mitochondrial transcription factor A (mtTFA/TFAM), and loading control Ponceau S from the rectus femoris muscle. Vertical dotted line demonstrates where images were cropped for representation. All samples were run on the same gel/membrane. Quantified relative expression of (**B**) MFN1, (**C**) MFN2, (**D**) OPA1, (**E**) FIS1, (**F**) DRP1, and (**G**) TFAM. (**H**) Relative gene expression of peroxisome proliferator- activated receptor gamma (PPARγ/PPARG) and PPARG co-activator 1 (PGC-1α/PPARGCA1). Unpaired *t*-test between C26 + 5FU and C26 + 5FU + Quer. n = 3*–*5/group. * *p* ≤ 0.05, ** *p* < 0.01.

**Figure 7 nutrients-15-00102-f007:**
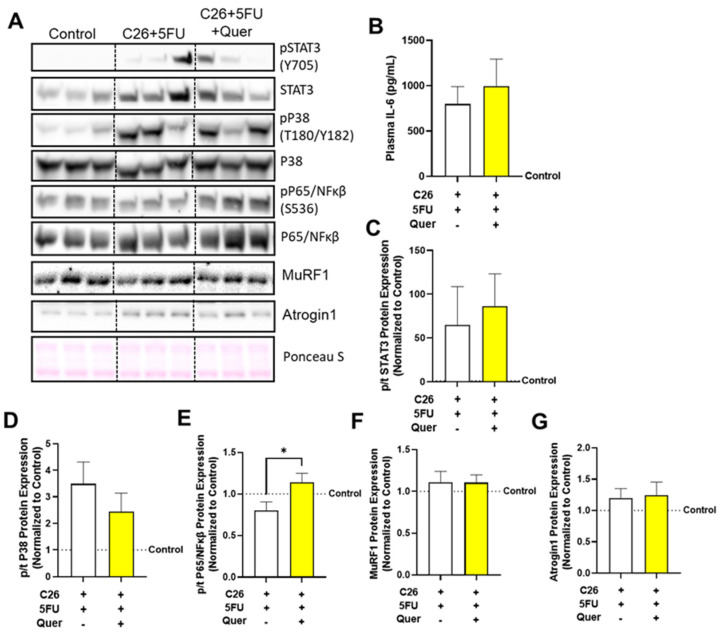
Quercetin has modest impact of circulating IL-6 and muscle inflammatory signaling. (**A**) representative Western blots of phosphorylated (p; tyrosine 705) and total (t) signal transducer and activator of transcription 3 (STAT3), phosphorylated (serine 536) and total P65, phosphorylated (threonine 180 and tyrosine 182) and total P38, muscle ring finger protein (MuRF) 1, atrogin 1, and loading control Ponceau S from the rectus femoris muscle. Vertical dotted line demonstrates where images were cropped for representation. All samples were run on the same gel/membrane. (**B**) Plasma interleukin 6 in picograms/milliliter (pg/mL). Quantified relative expression of (**C**) p/tSTAT3, (**D**) p/tP38, (**E**) p/tP65, (**F**) MuRF1, and (**G**) Atrogin1. Unpaired *t*-test between C26 + 5FU and C26 + 5FU + Quer. * *p* ≤ 0.05.

**Table 1 nutrients-15-00102-t001:** Antibodies and probes used.

Protein/Gene	Vendor	Catalog Number#
Total OXPHOS Cocktail	Abcam	Ab110413
VDAC	Cell Signaling Technology	4661
Cytochrome C	Cell Signaling Technology	11940
P62	Cell Signaling Technology	23214
LC3	Cell Signaling Technology	43566
Parkin	Cell Signaling Technology	2132
BNIP3	Cell Signaling Technology	3769
MFN1	Abcam	ab221661
MFN2	Cell Signaling Technology	9482
OPA1	Cell Signaling Technology	80471
FIS1	Abcam	ab229969
DRP1	Cell Signaling Technology	8570
TFAM	Abcam	ab252432
pSTAT3 (Y705)	Cell Signaling Technology	9145
STAT3	Cell Signaling Technology	4904
pP38 (T180/Y182)	Cell Signaling Technology	4511
P38	Cell Signaling Technology	8690
pP65/NFκβ (S536)	Cell Signaling Technology	3033
P65/NFκβ	Cell Signaling Technology	8242
MuRF1	Abcam	ab172479
Atrogin1	Abcam	ab168372
Anti-rabbit IgG -HRP linked	Cell Signaling Technology	7074
Anti-mouse IgG -HRP linked	Cell Signaling Technology	7076
*PPARGCA1*	Applied Biosystems	4351372
*PPARG*	Applied Biosystems	4331182

## Data Availability

Full uncropped western blots and histological images are available upon request.

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
