# Peer review of "Quercetin Improved Muscle Mass and Mitochondrial Content in a Murine Model of Cancer and Chemotherapy-Induced Cachexia"

_nutrients, 2022, doi:10.3390/nu15010102_

Round 1

Reviewer 1 Report

VanderVeen et al. investigated the effect of an organic compound Quercetin in treating cachexia induced by a combination of C26 tumours and 5-FU. The authors found that cachectic mice treated with Quercetin had improved mass and cross section area of some muscles. Increased muscle mitochondrial size and number were observed in Quercetin treated mice, which could be driven by improved mitochondrial homeostatic balance. While this study demonstrated a potential treatment for cancer cachexia in an animal model, the following points should be considered.

Major points:

1. As stated in the Materials and Methods section, the authors had four groups for the animal study: 1) Control (n=5), 2) C26 (n=5), 3) C26+5FU (n=5), 4) C26+5FU+Quer (n=5). What is the control group? Is it a non-tumour bearing group or a non-cachectic tumour group?  Body weight curve data from Control and C26 groups should also be included in Figure 1A.

2. The authors should present the recorded food intake data.

3. For figure 2A, data from the control group should be included, as the control group was used for comparison in other sub-figures.

4. Only mass of the rectus femoris and EDL muscle was significantly improved by Quercetin. However, the authors investigated the CSA and fibre sizes of TA muscles, although mass of TA muscles was not significantly different between groups. The authors should provide an explanation of why rectus femoris and EDL muscles were not investigated. Additionally, the authors should provide a discussion on why TA muscle weight was not different between C26+5FU and C26+5FU+Quer groups, while there were differences in CSA and fibre size.

5. It will be beneficial to investigate muscle expression of E3 ubiquitin ligases MuRF1 and Atrogin-1, which are markers indicating muscle wasting in the C26 model.

6. C26 model itself without 5-FU treatment is a classical murine model of cancer cachexia. Is there any reason why the author did not investigate the effect of Quercetin in the normal C26 model without 5-FU treatment?

7. Although Quercetin did increase mass of some muscles, it did not protect mass loss of other muscles such as Sol, Plant, Gas, and TA. It would be more precise to say that Quercetin partially protected against muscle loss.

Minor concerns

1. For Figure 1B, the average tumour mass in C26+5FU+Quer looks slightly higher than C26+5FU, though it is not statistically different. I suggest including individual data points in the figure.

2. For Figure 3 and 4, which muscle was used for the study?

3. For Figure 4A, it would be advantageous to have a blot with a longer exposure time.

4. For all western blots (Figure 4A, 5A, 6A, 7B), loading controls or ponceau red staining should be included to show equal loading.

5. Role of examined proteins such as VDAC, CytoC, BNIP3, MFN2, OPA1, FIS1, DRP1,TFAM, PPARG in mitochondria should be briefly explained to provide justification for why these proteins were studied.

Reviewer 2 Report

The manuscript by VanderVeen et al. describes a comprehensive experimental investigation on the effect of Quercetin in treating cachexia symptoms. The data is clearly presented, claims are well supported, control experiments are carried out properly, and the conclusions appear to be sound. I recommend publication with minor revision and include the following suggested improvement that the authors can consider:

  1. The discussion section, in its current form, is quite lengthy and rather repetitive to the introduction, the authors are encouraged to refine these paragraphs to highlight their findings.

Round 2

Reviewer 1 Report

The authors have appropriately addressed my concerns.